# Association of *IL-17A* and *IL-10* Polymorphisms with Juvenile Idiopathic Arthritis in Finnish Children

**DOI:** 10.3390/ijms25158323

**Published:** 2024-07-30

**Authors:** Milja Möttönen, Johanna Teräsjärvi, Heidi Rahikkala, Sonja Kvist, Jussi Mertsola, Qiushui He

**Affiliations:** 1Department of Paediatrics and Adolescent Medicine, Turku University Hospital, University of Turku, 20520 Turku, Finland; milja.mottonen@varha.fi (M.M.); heidi.rahikkala@varha.fi (H.R.); jusmer@utu.fi (J.M.); 2Institute of Biomedicine, Research Centre for Infections and Immunity, University of Turku, 20520 Turku, Finland; johter@utu.fi (J.T.); sonja.m.kvist@utu.fi (S.K.); 3Research Unit of Clinical Medicine, University of Oulu, 90014 Oulu, Finland; 4InFLAMES Research Flagship Centre, University of Turku, 20520 Turku, Finland

**Keywords:** juvenile idiopathic arthritis, JIA, SNP, IL-17A, IL-10, cytokine, treatment

## Abstract

To analyze the role of interleukin *IL-17A* and *IL-10* polymorphisms in susceptibility to juvenile idiopathic arthritis (JIA), 98 Finnish children and adolescents with JIA were studied. Data from the 1000 Genomes Project, consisting of 99 healthy Finns, served as the controls. The patients were analyzed for four *IL-17A* and three *IL-10* gene-promoter polymorphisms, and the serum IL-17A, IL-17F, IL-10, and IL-6 levels were determined. The *IL-17A* rs8193036 variant genotypes (CT/CC) were more common among the patients than controls, especially in those with polyarthritis (OR 1.93, 95% CI 1.11–3.36; *p* = 0.020). *IL-17A* rs2275913 minor allele A was more common in patients (OR 1.45, 95% Cl 1.08–1.94; *p* = 0.014) and especially among patients with oligoarthritis and polyarthritis than the controls (OR 1.61, 95%CI 1.06–2.43; *p* = 0.024). Carriers of the *IL-17A* rs4711998 variant genotype (AG/AA) had higher serum IL-17A levels than those with genotype GG. However, carriers of the variant genotypes of *IL-17A* rs9395767 and rs4711998 appeared to have higher IL-17F levels than those carrying wildtype. *IL-10* rs1800896 variant genotypes (TC/CC) were more abundant in patients than in the controls (OR 1.97, 95%CI 1.06–3.70; *p* = 0.042). Carriers of the *IL-10* rs1800896 variant genotypes had lower serum levels of IL-17F than those with wildtype. These data provide preliminary evidence of the roles of *IL-17* and *IL-10* in the pathogenesis of JIA and its subtypes in the Finnish population. However, the results should be interpreted with caution, as the number of subjects included in this study was limited.

## 1. Introduction

Juvenile idiopathic arthritis (JIA) is a pediatric rheumatic disease defined by the International League of Associations for Rheumatology (ILAR) as arthritis that has persisted for at least 6 weeks in one or more joints of a child or an adolescent under 16 years of age [1]. The ILAR classification further divides the disease into seven subtypes, including JIA with systemic onset, oligoarthritis (divided into persistent oligoarthritis and extended oligoarthritis that develops into polyarthritis after 6 months from onset), rheumatoid factor (RF)-positive polyarthritis, RF-negative polyarthritis, psoriatic arthritis, enthesis-related arthritis (ERA), and undifferentiated arthritis (not falling into any of previous categories or could be classified as more than one of the previous).

It is evident that JIA is a heterogenous group of immunological disturbances; its subtypes differ in clinical characteristics, disease activity, prognosis, and the most profitable treatment options [2]. Genes previously associated with JIA appear to be linked to a general susceptibility to autoimmunity rather than being specific to JIA [2,3]. It is likely that subtypes of JIA can arise from various genetic backgrounds which may differ between ethnic populations. Cytokines like interleukin (IL)-6, Tumor Necrosis Factor (TNF)-α, IL-1, IL-18, IL-17, and IL-10 play a crucial role in the pathogenesis of JIA, contributing to the inflammatory process and tissue damage [4].

Single-nucleotide polymorphisms (SNPs) in cytokine genes can lead to an altered expression of cytokines that appears to influence the risk of autoimmune diseases [5], including arthritis [6], their severity, and may also shape the responses to treatment. The assembly of polymorphisms of hallmark immune regulators may set up a context where an individual develops JIA, and these polymorphisms and their haplotypes may have a considerable influence on disease progression and response to treatment.

IL-17A and IL-17F produced by Th17 cells have been linked to the development and chronicity of synovial inflammation in rheumatic diseases, including JIA [7]. It has been suggested that these cytokines may be linked to certain subtypes of JIA and possibly disease activity in JIA [8,9]. Higher serum levels of IL-17A, among other inflammatory cytokines at JIA onset, have been shown to associate with ongoing disease activity after 1 year [10]. Also, the higher number of peripheral blood Th17 cells has been associated with a more prolonged time to reach an inactive disease state [11] and is associated with a prolonged need for treatment in polyarthritis [12]. Higher numbers of Th17 cells and lower numbers of Tregs are found in the joints of patients with extended oligoarthritis than in the joints of patients with persistent oligoarthritis, representing a more limited type of oligoarthritis [13,14]. Th17 cells appear to play an especially important role in ERA, where increased levels of Th17 cells and IL-17A levels have been found in synovial fluid (SF), correlating with disease activity [15]. Fischer et al. found the highest numbers of Th17 cells in the synovial fluid of patients with ERA and the lowest numbers in the joints of patients with antinuclear antibody (ANA)+ oligo- or polyarthritis [10]. In line with this, anti-IL-17A blocker secukinumab has been approved for the treatment of two subtypes of JIA: enthesitis-related arthritis and juvenile psoriatic arthritis.

IL-10 is a major anti-inflammatory cytokine that inhibits the activation and effector functions of T cells, the antigen-presenting cell function, and the proliferation of monocytes and macrophages [16]. It downregulates the production of TNF-α, IL-6, and other inflammatory cytokines and chemokines [17]. On the other hand, IL-10 induces B-cell proliferation, differentiation, and antibody isotype switching [11]. In animal models of arthritis, IL-10 deficiency has been demonstrated to result in an elevated production of Th17 and Th1 pro-inflammatory cytokines [18,19]. Genes associated with IL-10 signaling have been shown to be upregulated in the peripheral blood of patients with persistent oligoarthritis and seronegative polyarthritis, as well as in systemic JIA [14]. The plasma level of IL-10 as well as the level of IL-17 at diagnosis, have been listed among the predictive biomarkers of disease outcome in non-systemic JIA [20]. Some studies have found low levels of IL-10 in the joints or peripheral blood of patients with non-systemic JIA [21,22], and it has been suggested that intrinsic low IL-10 production insufficient to control inflammation may set up a risk of JIA with worse outcomes [23].

Several single-nucleotide polymorphisms (SNPs) in *IL-10* are identified across its coding and regulatory regions, which have been strongly implicated in the pathogenesis of various autoimmune diseases. In particular, three −1082 G>A (rs1800896), −819 C>T (rs1800871), and −592 C>A (rs1800872) promoter area mutations have been found to affect gene activity [24] and the secretion of IL-10 in the in vitro studies [25,26]. Previous studies have also shown that the *IL-10* ATA haplotype (minor alleles of rs1800896, rs1800871, and rs1800872) was associated with the reduced production of IL-10 in human whole blood cultures of patients with juvenile arthritis [23,27]. SNPs in the promoter region of *IL-17A* influence its expression and, potentially, JIA pathogenesis and outcome [28]. The most studied SNP in the *IL-17A* promoter region is rs2275913 (-197 G>A), which plays a pivotal role in the regulation of IL-17A transcription and the secretion of IL-17A [29]. In many studies, the A-allele is associated with elevated levels of IL-17A, however, findings are contradictory and, in some conditions, also reduced levels of IL-17A are reported.

In the era of biologicals, the clinical outcome of JIA has improved considerably [8,9], but a share of patients do not respond to treatment and develop complications. It remains necessary to advance our understanding of the disease mechanisms in JIA to better recognize the specific disease subtypes profiting from individual treatment approaches. In order to promote a more precise classification and targeted treatment of JIA, our approach has been to study whether the polymorphisms in *IL-17A* and *IL-10* genes are associated with susceptibility to JIA or its subtypes in a Finnish population and whether these polymorphisms could serve as biomarkers for more specific clinical entities of JIA. To our knowledge, comprehensive analyses of multiple SNPs in the promoter regions of *IL-17A* and *IL-10* genes have not been performed in JIA and its subtypes. In addition, contradictory results have been previously reported on the role of these polymorphisms with susceptibility to JIA from various genetic backgrounds. We chose to exclude the patients with sJIA and RF-positive polyarthritis from this study because of the rarity of these patients and the fact that these subtypes appear to differ markedly in the genetic background from the other subtypes of JIA [30]. 

## 2. Results

### 2.1. Characteristics of the Patient Population

Of the 98 patients with JIA included in the study, 71 (72.4%) were females, and 27 (27.6%) were males. All were of Caucasian origin, and all but one were of Finnish origin. At the time of blood sampling for the study, the median age of the patients was 11.5 (range 2.2–16.9) years, and the median disease duration was 2.9 (range 0–13.6) years. The median age at diagnosis was 5.4 years (range 1.0–14.9). Fifty-one (52.0%) patients were diagnosed as having oligoarthritis, thirty-four (34.7%) with polyarthritis, ten (10.2%) with enthesis-related arthritis (ERA), and three (3.0%) had other types of JIA. Twenty-nine patients (29.9%) were human lymphocyte antigen B27 (HLA-B27) positive and sixty-five (67.7%) were ANA-positive. A total of 2 patients (20.4%) had uveitis at some point in the disease course, and 26 (26.5%) patients had temporomandibular joint (TMJ) synovitis at some point in the disease course. Detailed characteristics of the JIA patients are shown in Table 1. 

Next, we compared the prevalence of *IL-17* and *IL-10* polymorphisms between the control population and patients with JIA. In both groups, no significant sex-based differences were observed concerning the prevalence of SNPs (*p* ≥ 0.05).

### 2.2. IL-17A Polymorphisms

The distribution of *IL-17A* polymorphisms rs9395767, rs2275913, rs4711998, and rs8193036 were analyzed in JIA patients and the control group (Table 2). As shown in Table 2, *IL-17A* rs8193036 variant genotypes CC/CT were more common than the wildtype genotype TT in patients than the controls (71.4% vs. 59.6%, *p* = 0.038, OR 1.92; 95% Cl 1.06–3.47). Although, the minor allele A of *IL-17A* rs2275913 was more frequent in the JIA patients than the controls (OR 1.45, 95% Cl 1.08–1.94; *p* = 0.014) 

Subsequently, the patients with oligoarthritis and polyarthritis were analyzed as a group (*n* = 85) against the controls (Table 3). What stood out was that both the *IL-17A* rs2275913 variant genotype GA/AA and *IL-17A* rs8193036 variant genotype CT/CC were significantly more common in the patient group than in the controls (*p* = 0.043 odds ratio (OR) 2.0; 95% confidence interval (Cl) 1.02–3.93 and *p* = 0.007, OR 2.11; 95% Cl 1.2–3.6, respectively). In this combined group of oligoarthritis and polyarthritis, *IL-17A* rs2275913 minor allele A was more common than in the control group (*p* = 0.023 (OR 1.59; 95% C1 1.07–2.37). Interestingly, *IL-17A* rs8193036 C-allele was more frequent in the polyarthritis group than in the controls (*p* = 0.020, OR 1.93; 95% Cl 1.11–3.36).

In a further analysis where the patients with extended oligoarthritis (*n* = 7) were excluded, and the patients with persistent oligoarthritis (*n* = 44) and polyarthritis (*n* = 34) were compared, no statistical differences in the distribution of *IL-17A* polymorphisms were found. However, when the patients with persistent oligoarthritis were compared to the patients with extended oligoarthritis, we found that all the patients with the extended type of oligoarthritis had the variant types of both *IL-17A* rs2275913 and rs8193036 genotypes, whereas 20.5% of the patients with persistent oligoarthritis had the wildtype (GG) of *IL-17A* rs2275913 and 34% had the wildtype (CC) of *IL-17A* rs8193036. However, the number of patients with extended oligoarthritis was too low to draw definite conclusions about the differences (Appendix A).

As shown in Table 4, the majority (82.4%) of JIA patients who carry the *IL-17A* rs9395767 TT-genotype were ANA positive, whereas only 55.6% of those who carry the AA genotype were ANA positive (*p* = 0.032, OR 3.8; 95% Cl 1.13–13.09). No significant correlation between the distribution of other *IL-17A* polymorphisms and ANA-positivity were observed.

We further studied the link between *IL-17A* polymorphisms with the risk of uveitis and the risk of a typical feature of JIA: TMJ arthritis. The JIA patients were divided into those who had uveitis at some point in the disease course (*n* = 20) and into a group that never had uveitis (*n* = 78). No statistically significant differences were found in the studied IL-17A genotypes between the patients with a positive vs. negative history of uveitis when the whole population of JIA was analyzed (Table 4). Furthermore, there were no significant differences when the patients with oligoarthritis, polyarthritis, and ERA were analyzed separately. A total of 26 patients (26.5%) had been TMJ-affected at some point in the disease course. No correlations were found either when we similarly studied the association of *IL-17A* polymorphisms with a history of TMJ arthritis in these patient groups (Table 4).

### 2.3. IL-10 Polymorphisms

The distributions of three *IL-10* polymorphisms (rs1800896, rs1800871, and rs1800872) were analyzed in JIA patients and the control group (Table 2). *IL-10* rs1800896 variant genotypes (TC/CC) were significantly more abundant in the JIA group than in the controls (*p* = 0.042, OR 1.66; 95%Cl 1.02–2.72). No difference in the other two SNPs wasnoticed between the patients and controls. Furthermore, no difference in all three SNPs studied was found between the different subtypes of JIA studied (Table 3). Nevertheless, variant *IL-10* rs1800896 genotype may be more abundant in ERA than in the controls. Of the ten patients with ERA, nine had the variant genotype. However, the number of ERA patients is too low to draw definitive conclusions. Similarly, no associations between the *IL-10* polymorphisms and patients with oligoarthritis, polyarthritis, or the combined group of oligoarthritis and polyarthritis were found.

No statistically significant differences were found between the *IL-10* rs1800896, rs1800871, and rs1800872 genotypes in patients with positive vs. negative ANA (Table 4). However, when the JIA patients were analyzed against the controls, there seemed to be an association with the *IL-10* rs1800896 variant genotype GA/AA and HLAB27 positivity (*p* = 0.033; OR 3.01; 95% CI 1.04–8.71). Furthermore, no statistically significant differences were found in the *IL-10* rs1800896, rs1800871, and rs1800872 genotypes in patients with a positive vs. negative history of uveitis or a positive or negative history of TMJ arthritis when the whole population of JIA patients and the different subtypes were analyzed.

### 2.4. IL-10 and IL-17 Haplotypes

The linkage disequilibrium (LD) of *IL-10* polymorphisms was strong in the controls (Figure 1a) and the JIA patient group (Figure 1b). In addition, *IL-10* rs1800872 and rs1800871 exhibit complete LD. There was no evidence of LD in the IL-17A SNPs in the JIA group, and in the control group, only *IL-17A* rs2275913 was low, evidenced by the LD with rs8193036 and rs9395567. The haplotype analysis showed that the *IL-17A* rs2275913 and rs819306 allelic TG-haplotype was more frequent in the control group (0.433) than in the patient group (0.321) (*p* = 0.0211). No other significant differences were found in the *IL-17* and *IL-10* haplotypes between the patients and controls.

Next, we analyzed the LD of three JIA subgroups: oligoarthritis, polyarthritis, and ERA. What stands out in Figure 2 is that the ERA group clearly differed in terms of the IL-10 haplotypes, both in the controls and the other JIA subgroups. The most significant difference was related to *IL-10* rs1800896 and rs1800872/rs1800871, where there was high evidence of LD between the SNPs in the polyarthritis group (Figure 2b) but not in the oligoarthritis group (Figure 2a). 

### 2.5. The Role of IL-10 and IL-17 Polymorphisms in Disease Activity

To study whether the *IL-17A* and *IL-10* polymorphisms were associated with disease activity, the patients with oligoarthritis or polyarthritis were divided into four groups (inactive disease, low disease activity, medium disease activity, and high disease activity) on the basis of their JADAS10 scores, as defined by Consolaro et al. [33,34], at two different time points: at diagnosis and at 1 year after diagnosis. 

At the time of diagnosis, all patients had active disease, with 98.3% having moderate (*n* = 13) or high (*n* = 44) disease activity (Figure 3). Most of the patients with moderate disease activity at the time of diagnosis had persistent oligoarthritis. One year after diagnosis, 32.4% (*n* = 24) of patients had inactive disease, and 27% (*n* = 20) had low disease activity. Despite active treatment, one year after diagnosis, 28.4% (*n* = 21) of patients had moderate and 12.2% (*n* = 16) high disease activity (Figure 3). While the patients diagnosed with poly- or extended oligoarthritis seemed to have more active disease one year after diagnosis, the patients with polyarthritis also showed a more systematic reduction in disease activity compared to the time of diagnosis compared to other subtypes. 

The polymorphisms of *IL-17A* (rs8193036, rs2275913, rs9395767, and rs4711998) and *IL-10* (rs1800896 and rs1800871) and their association with disease activity were analyzed using the collected juvenile arthritis disease activity score (JADAS10), physician’s global assessment (PGA) and the childhood health assessment questionnaire (CHAQ) data. It appeared that the *IL-10* rs1800871 variant genotype may be associated with disease activity. No statistically significant differences in JADAS10 scores at diagnosis or 1 year after were observed between patients with the WT or variant rs1800871 genotype, but at one year after diagnosis, the PGA score was significantly lower (median 0.00 IQR 0.5) in patients with the variant genotype vs. patients with the WT genotype (median 0.6 IQR 1.38, *p* = 0.026). Also, the JIA patients carrying the variant type of *IL-10* rs1800871 had a significantly lower CHAQ at the time of diagnosis (0.25; IQR 0.59, *p* = 0.013) and one year after (0.00; IQR 0.25, *p* = 0.01) compared with the WT carriers. No significant associations were found between disease activity and the other studied SNPs. 

### 2.6. The Effect of IL-17A and IL-10 Polymorphisms on Serum Cytokine Levels

To analyze the functional significance of the *IL-17A* and *IL-10* polymorphisms in JIA, we measured the concentrations of IL-6, IL-10, IL-17A, and IL-17F in the sera of the patients and compared them to the genotypes. The mean concentrations of serum cytokine levels are shown in Table 5. 

Pearson’s correlation analysis of the cytokine levels showed a positive correlation between the serum IL-17F and IL-10 levels (r^2^ = 0.599, *p* ≤ 0.001). No correlation was observed between the other cytokine levels. No statistically significant differences were found in the cytokine levels between the sexes, albeit males (22.61 pg/mL, interquartile range (IQR) 49.39) appeared to have higher serum IL-10 levels compared to females (5.11 pg/mL, IQR 44.74) (*p* = 0.093). 

As seen in Table 5, of the three subgroups presented, polyarthritis patients had the highest concentration of serum IL-6 (264.24 pg/mL; IQR 9706.65) and ERA patients the highest concentration of serum IL-10 (30.87 pg/mL; IQR 51.27). However, no statistically significant differences were observed in the serum cytokine levels across the categories of diagnoses. The only significant difference found was between oligo- and polyarthritis patients, with the polyarthritis patients having higher IL-17A levels (*p* = 0.049).

The patients with oligo- or polyarthritis were further divided into four activity categories, as described previously. Interestingly, the levels of IL-17F (*p* = 0.012) and IL-10 (0.006) differentiated across the categories of disease activity, being lowest in the inactive patients and highest in those with moderate disease activity. However, incoherently, the patients in the high-activity group had very low levels of all studied cytokines. We did not observe the association of serum levels of IL-17A and IL-6 with disease activity.

When the associations between individual polymorphisms of IL-17A and IL-10 and the cytokine levels were further analyzed, *IL-17A* rs9395767 and rs4711998 and *IL-10* rs1800896 were found to be associated with the serum IL-17F levels. Carriers of the variant type of the above-mentioned *IL-17* SNPs had significantly higher levels of serum IL-17F than those who had wild genotypes (Table 6). In addition, the *IL-17A* rs4711998 variant genotype carriers had higher serum IL-17A levels than the WT carriers when the dominant model (combined heterozygote and homozygote variants are compared with the WT) was used (*p* = 0.019). Opposite to these findings, JIA patients who carry the variant type of *IL-10* rs1800896 had a lower level of serum IL-17F (*p* = 0.010, with the dominant model). None of the studied SNPs appeared to be statistically significantly associated with either the serum IL-10 or IL-6 levels.

The Quade nonparametric ANCOVA test was used to take account of the covariates as listed: sex, JIA subtype, age at the time of sampling, disease duration, JADAS10, and the use of biological medication. The test clearly showed that *IL-17A* rs471198 was associated with serum IL-17F (*p* = 0.012) and IL-17A (*p* = 0.009) levels. *IL-10* rs1800896, on the other hand, was associated with both serum IL-17F (*p* = 0.030) and IL-10 (*p* = 0.042) levels.

## 3. Discussion

In this exploratory study, we aimed to study the role of several *IL-17A* and *IL-10* promoter area gene polymorphisms in the susceptibility to and disease activity of JIA in children. It is known that SNPs in cytokine genes affect cytokine production, which can influence the risk of arthritis [23,35]. To our knowledge, this is the first study addressing the role of *IL-17A* and *IL-10* polymorphisms in JIA in the Finnish population. Disease characteristics of the patients in this study, including the distribution of JIA subtypes, age, and gender, are in line with what has previously been reported in Nordic studies [36,37]. Patients with sJIA and seropositive polyarthritis were not included in the study due to the low prevalence of these subtypes and differing pathogenesis from other subtypes. This study is representative of the roles of *IL-17A* and *IL-10* polymorphisms, mainly in the oligoarthritis and polyarthritis types of JIA (which are the two major groups) in a Finnish population, as the numbers of ERA patients and other types of arthritis were low.

The *IL-17A* rs8193036 and rs2275913 variant genotypes were found to increase the risk of JIA in our cohort. IL-17A rs2275913 variant genotypes seem to be associated with an increased risk of RA in Caucasians [38,39], but in a study by Zhang et al., no association with the risk of JIA in Chinese children was observed [40]. The *IL-17A* rs8193036 variant genotype has been previously associated with an increased risk of rheumatoid arthritis (RA) in the Chinese population [41]. However, a meta-analysis, including fourteen studies and 3118 patients with RA, did not find an association between *IL-17A* rs8193036 and susceptibility to RA [42]. Our analyses confirm that the *IL-17A* rs8193036 polymorphism did not significantly affect the serum levels of IL-17A or other cytokines. However, it should be kept in mind that the majority of patients in our study were receiving medication, including DMARDS and biologicals, at the time of sampling, which may affect the results; although, our data suggest that the cytokine levels were not dependent on the use of biological drugs. 

Our results suggest that both *IL-17A* rs8193036 variant genotypes CT/CC and *IL-17A* rs2275913 variant genotypes GA/AA increase the risk of oligoarthritis and polyarthritis types of JIA. Interestingly, our results showed that *IL-17A* rs8193036 minor allele C is associated with an increased risk, especially for the polyarthritis type of disease. Extended oligoarthritis has basically the same disease features as polyarthritis but with a more gradual onset, and in line with this, among the oligoarthritis group, all patients with an extended type of oligoarthritis had the variant genotype type of both *IL-17A* rs8193036 and *IL-17A* rs2275913. The number of patients with extended arthritis was low, but it remains of interest to further study if the *IL-17A* rs8193036 polymorphism in oligoarthritis patients could serve to predict the risk of disease extension. Thus far, there has been no published data on utilizing secukinumab (currently the only IL-17 blocker having an indication in pediatric rheumatic disease) in the treatment of other pediatric rheumatic conditions than ERA and psoriatic arthritis. It is tempting to speculate that some oligoarthritis or polyarthritis patients with the *IL-17A* rs8193036 variant genotype could benefit from IL-17 blocking.

ANA-positive patients have been suggested to form a subtype of JIA independent of the number of joints affected [30]. Typical features of ANA-positive JIA include female predominance, disease onset in the early years, and a high risk of uveitis [30]. However, neither *IL-17A* rs8193036 nor *IL-17A* rs2275913 seemed to associate with ANA-positivity in our study. In contrast, the *IL-17A* rs9395767 variant genotype was more abundant in ANA-positive patients than in ANA-negative patients, with the majority of patients with the variant genotype being ANA-positive. The functional role of this particular *IL-17A* SNP remains uncharacterized. A study by Gan et al. has demonstrated that higher serum levels of IL-17A, and especially higher levels of IL-17F, are associated with higher autoantibody (including ANA) levels in patients with primary Sjögren’s syndrome. Interestingly, treatment of patients with psoriasis or psoriatic arthritis with IL-17 blocker secukinumab has been shown to lead to diminished levels of ANA [43]. None of the studied *IL-17A* or *IL-10* polymorphisms seemed to mediate the risk of uveitis. However, uveitis is not always present at disease onset and the risk of developing uveitis is greatly diminished after disease onset while patients are receiving efficient therapy, which may influence the outcome in this type of study setting.

*IL-10* polymorphisms have been previously associated with the risk of JIA in some studies. Fathy et al. (2017) have demonstrated an increased risk of JIA and, especially, of polyarthritis in Egyptian children with the *IL-10* rs1800896 variant AA genotype [44]. However, no association of *IL-10* rs1800896, *IL-10* rs1800871, and *IL-10* rs1800872 with JIA was found in an Iranian study [45], nor in the meta-analysis that included seven studies by Jung et al. [46]. In our study, *IL-10* polymorphisms rs1800896, rs1800871 and rs1800872 were not found to significantly associate with the risk of JIA or its subtypes in this Finnish JIA population. However, the *IL-10* rs18000896 variant/*IL-10* rs1800871 variant haplotype was more common in the JIA group compared to the controls, suggesting that while the effect of one individual variation in the *IL-10* gene promoter is not enough to increase the risk of JIA, two might be. Based on most studies on the *IL-10* rs 18000896 polymorphism in chronic inflammatory diseases, it is commonly believed to lead to diminished IL-10 production, although some studies have associated it with higher IL-10 production [47]. In our study, both the *IL-10* 18000896 variant genotype and *IL-10* rs 18000896 variant/*IL-10* rs 1800871 variant haplotype were associated with lower IL-10 production. Also, Fathy et al. have found lower serum levels of IL-10 in JIA patients with the variant *IL-10* rs1800896 genotype compared to those with the WT genotype [44]. A study by Hee et al. (2007) investigating the role of the *IL-10* gene promoter polymorphism in RA in Malaysian patients showed that the haplotype comprising all minor alleles in rs1800896, rs1800871, and rs1800872 (ATA haplotype) was associated with lower IL-10 production when compared with the other haplotypes, and the RA patients who did not display the ATA haplotype produced significantly higher levels of IL-10 when compared with those carrying either one or two polymorphisms [48].

Based on PGA, patients with the *IL-10* rs1800871 variant genotype were found to have significantly lower disease activity 1 year after diagnosis compared to patients with the *IL-10* rs1800871 WT genotype. This could reflect a better response to medication, as there was a trend towards higher PGA with the patients with variant genotypes at diagnosis; however, this data did not reach statistical significance. CHAQ was lower in patients with the *IL-10* rs1800871 variant genotype both at diagnosis and at 1 year. However, the correlation of CHAQ with the joint counts has been shown to be low in early disease and better with more longstanding disease, with the correlation increasing alongside disease duration [49]. In a previous work by Schotte et al., the association of *IL-10* promoter SNPs (−2849 G>A (rs6703630), −1082 G>A (rs1800896), −819 C>T (rs1800871), and −592 C>A (rs1800872) with a response to etanercept treatment in RA patients was studied [50]. They found the most favorable response in patients with the -2849 A-allele or the haplotypes AGCC and GATA, whereas an unfavorable treatment response was found in patients with the GGCC genotype [50]. 

Although the number of ERA patients included in our study was very low, it stands out that IL-10 is differently regulated in ERA compared to oligoarthritis and polyarthritis. Our results indicate a trend towards the association of ERA with the variant *IL-10* rs1800896 genotype, which would be in line with the findings by Braga et al., who have shown that the rs1800896 variant genotype increased the risk of ankylosing spondylitis (AS) by three-fold [51]. They also showed that this association was independent of HLA-B27. Also, a study by Mu et al. recognized *IL-10* rs1800896 (along with other IL-10 polymorphisms) as a risk factor for AS in the Chinese population [52]. The *IL-10* rs1800896 variant genotype has been shown to be associated in a number of studies with high *IL-10* production in AS [51]. While we did not find increased serum levels of IL-10 in all JIA patients with the *IL-10* rs1800896 variant genotype, rather, we found decreased serum levels along with decreased IL-17F levels compared to patients with the WT genotype; however, this may not apply to patients with ERA. Furthermore, there was a general trend towards higher IL-10 serum levels in these patients. 

The global incidence of juvenile idiopathic arthritis (JIA) is relatively low. For instance, in Finland, the incidence rate is 31.7 cases per 100,000 individuals [53]. This low incidence rate significantly hampers the recruitment process, resulting in a slower accrual of study participants. Consequently, the limited number of available patients represents a primary limitation of this study, and in the case of ERA patients, the number of subjects was definitely too low to reliably define the population. 

Therefore, we considered our study to be an exploratory study to discover the preliminary associations of IL-17A and IL-10 polymorphisms with JIA, and, if present, they need to be replicated before more credence is given to these results. Specially, the correlations of polymorphisms with disease activity need to be confirmed in larger populations. Another clear limitation regarding the cytokine data is that the majority of patients were on varying medications at the time of sampling and that this most likely affected the results; however, the medication was included in the Quade nonparametric ANCOVA model as one covariate to minimize an effect.

## 4. Material and Methods

### 4.1. Study Design and Subjects

All study subjects were recruited from the Pediatric Rheumatology Clinic at Turku University Hospital in Turku, Finland, between November 2020 and September 2023. Altogether, 130 patients were invited, of which 122 provided their consent to participate and were enrolled. Study samples were collected by providing the guardian with coded tubes to be given to the lab personnel while having routine laboratory tests taken and, eventually, study samples were obtained from 98 patients who were included in the analyses. At the time of initial recruitment, all patients were under 16 years of age and fulfilled the ILAR classification criteria for JIA. The exclusion criteria included systemic onset of JIA, seropositive polyarthritis (due to the expected small number of patients), a known chromosomal abnormality, known genetic disorder, participation in other clinical intervention studies, or whose informed consent could not be ensured due to the lack of a shared language or other factors. As a control population, we used genetic data from the 1000 Genomes Project, which consists of 99 healthy Finns (FIN population) [54]. In addition, we analyzed one SNP of *IL-17* (rs2275913) and two SNPs of *IL-10* (rs1800896 and rs1800871) in our previous STEP study in which healthy infants were followed [55].

Before enrollment in the study, the patients and their guardians were informed about the procedures and the aim of the study, and their informed written consent was obtained. The current study was approved by the Ethics Committee of Turku University and the Hospital District of Southwest Finland (ETMK 31/1801/2020, 16 June 2020).

The following clinical data/parameters were collected: the age at disease onset (time of diagnosis and birthdate), the age at the time of sampling, sex, duration of disease, erythrocyte sedimentation rate (ESR), C-reactive protein (CRP) levels, antinuclear antibody titer (ANA), Human Leucocyte Antigen (HLA)-B27, number of affected joints, history of uveitis, history of tempo mandibular joint (TMJ) arthritis, medication, child health assessment questionnaire (CHAQ), physician global assessment of disease activity (PGA) and juvenile arthritis disease activity score (JADAS10). The cumulative activity and clinical parameters at the selected time points (at diagnosis, one year after diagnosis, three years after diagnosis, at the start of the first biological, one year after the start of the first biological, and at the time of sampling) were retrospectively collected from the medical records by a physician. 

### 4.2. Sampling and Data Collection

Serum and EDTA samples were drawn from a peripheral vein at the time of routine monitoring tests. The serum samples were allowed to clot for 60 min at room temperature and were then centrifuged at 2000× *g* for 10 min at 4 °C, transferred into two cryovials, and stored at −20 °C until used for the analyses. 

### 4.3. Genetic Analyses

Genomic DNA was extracted from 250 μL of EDTA whole blood samples using the E.Z.N.A Blood DNA Mini Kit (Omega Bio-tek, Inc., Norcross, GA, USA) according to the manufacturer’s protocol. DNA concentrations were determined by a spectrophotometer (NanoDrop 2000, Thermo Scientific, Waltham, MA, USA), and the DNA samples were stored at −20 °C prior to the analyses.

A total of eight *IL-17* and three *IL-10* polymorphisms were analyzed (Figure 4) from JIA patients using Sanger sequencing at Eurofins Genomics (Eurofins Genomics GmbH, Konstanz, Germany). All the SNPs that had a minor allele frequency of >5% were included in the final analyses and presented in Table 2.

The primers for *IL-17* and *IL-10* were designed with the Primer-BLAST design tool (National Center for Biotechnology Information (NCBI), U.S. National Library of Medicine, Bethesda, MD, USA) and were ordered from Eurofins Genomics (Eurofins Genomics GmbH, Konstanz, Germany). The Invitrogen Platinum Taq DNA polymerase (Thermo Fisher Scienific Inc., Waltham, MA, USA) was used for the PCR reactions according to the manufacturer’s instructions. All PCR reactions were performed in a total volume of 30 µL, consisting of 4 µL of genomic DNA and 26 µL of master mix, including 1.5 mM MgCl2, 1× buffer, 0.2 mM dNTP′s, 0.2 µM of each primer, and 2U Platinium Taq DNA polymerase enzyme (Thermo Fisher Scientific). The PCR conditions were as follows: an initial activation of 2 min at 95 °C and for 40 cycles of repeated denaturation at 94 °C for 30 s, annealing at the primer-specific temperature for 30 s, and extension at 72 °C for 1 min and without the final extension step. The primers and annealing temperatures used in the PCR are listed in Table 7. Prior to the sequencing, the PCR products were purified enzymatically with Thermo Scientific Exonuclease FastAP and Exo I (Thermo Fisher Scientific, Waltham, MA, USA).

*IL-17A* rs2275913 and IL-10 rs1800896 and rs1800871 SNPs from the STEP study subjects have been previously analyzed using the Sequenom massARRAY iPlex Gold system (Sequenom Inc., San Diego, CA, USA) at the University of Eastern Finland, Kuopio, Finland [55].

### 4.4. Cytokine Measurements

The serum cytokine concentrations of IL-17A, IL-17F, IL-10, and IL-6 were measured using the multiplex immunoassay (Bio-PLex 200, Bio-Rad Laboratories, Hercules, CA, USA) with the Milliplex Map Human Th17 Magnetic Bead Panel (Merck KgA, Darmstadt, Germany) according to the manufacturer’s protocol. There was no or negligible cross-reactivity between the antibodies for an analyte or any of the other analytes in the panel. The reported intra-assay %CVs were 3, 2, 3, and 5, and the inter-assay %CVs were 13, 10, 11, and 7 for IL-17A, IL-17F, IL-10, and IL-6, respectively. The cytokine concentrations that were below the lowest detection limit were assigned values of half of the minimum detectable concentration of each cytokine, which were 1.05 pg/mL, 4.5 pg/mL, 0.15 pg/mL, and 0.85 pg/mL for IL-17A, IL-17F, IL-10, and IL-6, respectively. 

### 4.5. Statistical Analysis

Statistical analyses were performed using SPSS software, version 28.0 (IBM Corp. in Armonk, NY, USA) and GraphPad Prism, version 8 (GraphPad Software, La Jolla, MA, USA). The calculation of the sample size and power analysis were performed with the QUANTO program, version 1.2.4 [56]. The current sample size would have 35–93% power at a 0.05 significance level to detect a difference between the controls and patients with JIA-type when the controls per case are 1:10 (IL-17A rs2275913 and IL-10 rs1800896, rs1800871, and rs1800872). When the control per case is 1:1, the power of the study is 57% at a 0.05 significance level, indicating that these results are exploratory and should be interpreted with a critical perspective. 

Deviations from the Hardy–Weinberg equilibrium (HWE) for the *IL-17* and *IL-10* SNPs were studied using the Chi-square test. Categorical data were compared by using the Chi-square test or Fisher exact test. The non-normally distributed data were compared using the Mann–Whitney U test. Odds ratios (OR) with 95% CIs were determined, and repeated measures ANOVA was used to analyze the repeated measurements of the subjects. The Quade nonparametric ANCOVA test was used to calculate the associations between serum IL levels and the SNPs with the following covariates: sex, JIA subtype, age at the time of sampling, disease duration, JADAS10 (= disease activity), and the use of biological medication. A two-tailed *p* < 0.05 was considered significant, and in the haplotype analyses, the Bonferroni correction was used to determine the adjusted *p*-values. 

## 5. Conclusions

In this study, we present new evidence on the roles of *IL-17A* and *IL-10* polymorphisms in the pathogenesis of JIA. We have shown that carrying *IL-17A* rs2275913 (AG/AA) or rs8193036 variant genotypes (CT/CC) and *IL-10* rs1800896 variant genotypes (TC/CC) increases the risk of JIA, and specifically, the risk of seronegative polyarthritis is increased in the *IL-17A* rs8193036 variant background. Also, *IL-17A* rs2275913 minor allele A was identified as a risk factor for oligoarthritis and polyarthritis types of JIA. We further show that the *IL-10* rs1800871 variant genotype may be related to a better response to treatment at 1 year. Differences in the serum cytokine profiles between the patients with wildtype and variant genotypes of *IL-17A* and *IL-10* polymorphisms were detected. Further studies with treatment of naïve patients are needed to specify the associations with cytokine production. Also, further studies with larger sample sizes are planned so as to specify the role of these cytokine polymorphisms in disease subtypes and patients’ responses to treatment.

## Figures and Tables

**Figure 1 ijms-25-08323-f001:**
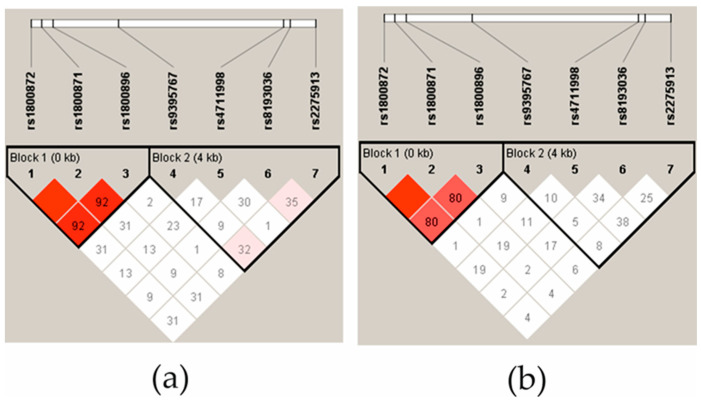
The linkage disequilibrium (LD) block structure of controls (**a**) and JIA patients (**b**). LD consisted of four SNPs located in the IL-17A gene and three SNPs located in the IL-10 gene. The LD is displayed according to following colour schemes, with bright red: LOD > 2, D′ = 1, shades of pink/red: LOD > 2, D′ < 1, D′ = 1 and white: LOD < 2, D′ < 1. D’ values multiplied by 100 are marked in each cell, cells without number indicates D′ 100.

**Figure 2 ijms-25-08323-f002:**
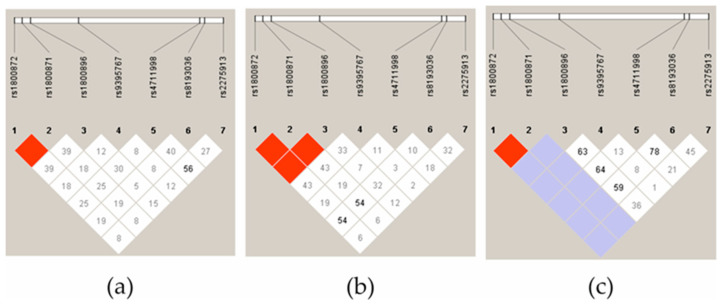
The linkage disequilibrium (LD) block structure consisted of four SNPs located in the IL-17A gene and three SNPs located in the IL-10 gene in patients with oligoarthritis (**a**), polyarthritis (**b**), and ERA (**c**). The LD is displayed according to following colour schemes, with bright red: LOD > 2, D′ = 1, shades of pink/red: LOD > 2, D′ < 1, blue: LOD < 2, D′ = 1 and white: LOD < 2, D′ < 1. D’ values multiplied by 100 are marked in each cell, cells without number indicates D′ 100.

**Figure 3 ijms-25-08323-f003:**
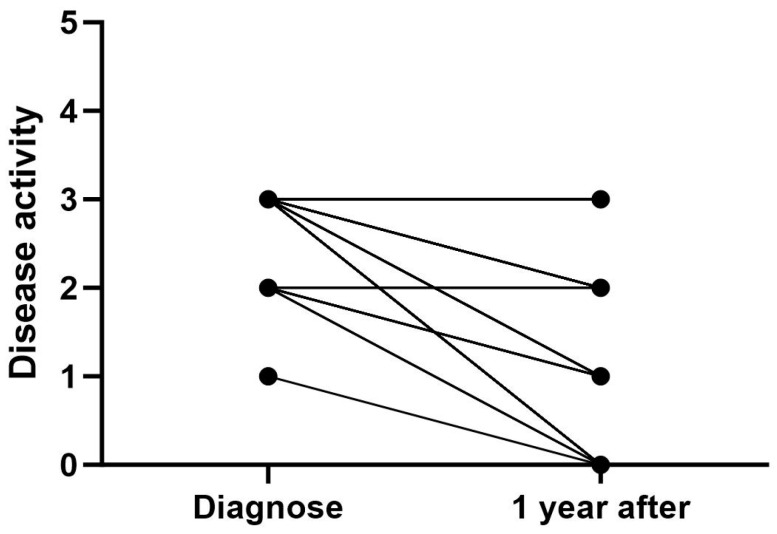
Disease activity in all JIA patients at the time of diagnosis and 1 year after diagnosis. Disease activity score: 0 = inactive, 1 = low activity, 2 = moderate activity, and 3 = high activity.

**Figure 4 ijms-25-08323-f004:**
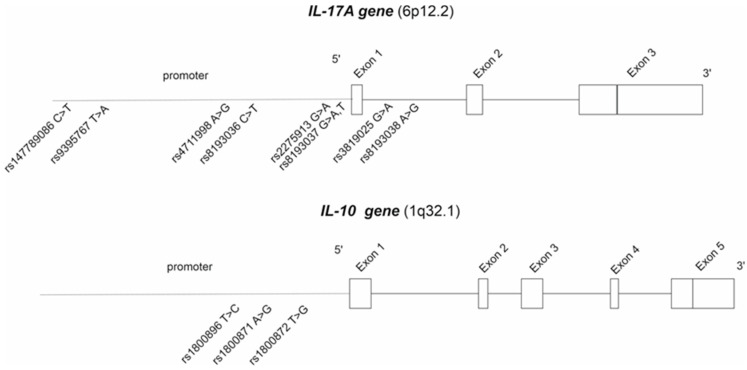
The analyzed IL-17A and IL-10 polymorphisms.

**Table 1 ijms-25-08323-t001:** Characteristics of the JIA patients.

	Juvenile Idiopathic Arthritis (JIA) Subtypes				
Parameter	All Patients, *n* = 98 (%)	Oligo Arthritis ^1^, *n* = 51 (%)	Polyarthritis, *n* = 34 (%)	ERA, *n* = 10 (%)	Other ^2^, *n* = 3 (%)
**Sex**					
Female	71 (72.4)	38 (74.5)	29 (85.3)	1 (10.0)	3 (100.0)
Male	27 (27.6)	13 (25.5)	5 (14.7)	9 (90.0)	0 (0.0)
Age of Diagnose (median, min – max, years)	5.4 (1.0–14.9)	5.3 (1.4–13.7)	3.9 (1.0–14.9)	11.4 (6.8–14.6)	5.9 (3.7–12.6)
Disease duration at time of sampling (min–max, years)	2.9 (0–13.6)	3.4 (0.0–13.6)	3.9 (0.0–12.5)	1.4 (0.2–4.2)	1.0 (0.0–1.8)
**Medication ^3^**					
DMAR only	34 (34.7)	18 (15.7)	10 (29.4)	4 (40.0)	2 (66.7)
Biologics only	18 (18.4)	9 (17.6)	9 (26.5)	0 (0.0)	0 (0.0)
DMARD and biologics	34 (34.7)	16 (31.4)	14 (41.2)	4 (40.0)	0 (0.0)
No medication	12 (12.2)	8 (15.7)	1 (2.9)	2 (20.0)	1 (33.3)
**Disease activity** ^4,5^					
Inactive (A)	0 (0.0)	0 (0.0)	0 (0.0)	-	-
Low activity (A)	1 (1.7)	1 (3.0)	0 (0.0)	-	-
Moderate activity(A)	13 (22.4)	3 (9.1)	10 (40.0)	-	-
High activity (A)	44 (75.9)	29 (87.9)	15 (60.0)	-	-
Missing data	40 (40.8)	18	9		
Inactive (B)	24 (32.4)	14 (31.8)	10 (33.3)	-	-
Low activity (B)	20 (27.0)	9 (20.5)	11 (36.7)	-	-
Moderate activity (B)	21 (28.4)	13 (29.5)	8 (26.7)	-	-
High activity (B)	9 (12.2)	8 (18.2)	1 (3.3)	-	-
Missing data	24 (24.5)	7	4		
Inactive (C)	42 (53.2)	24 (53.3)	18 (52.9)	-	-
Low activity (C)	7 (8.9)	4 (8.9)	3 (8.8)	-	-
Moderate activity (C)	14 (17.7)	7 (15.6)	7 (20.6)	-	-
High activity (C)	16 (20.3)	10 (22.2)	6 (17.6)	-	-
Missing data	19 (19.4)	6	0		
**HLA-B27**					
Positive	29 (29.9)	9 (17.6)	11 (33.3)	9 (90.0)	0 (0.0)
Negative	68 (70.1)	42 (82.4)	22 (66.7)	1 (10.0)	3 (100)
Missing data	1 (1.1)	-	1 (3.1)	-	
**ANA**					
Positive	65 (67.7)	35 (70.0)	26 (76.5)	2 (22.2)	2 (66.7)
Negative	31 (32.3)	15 (30.0)	8 (23.5)	7 (77.8)	1 (33.3)
Missing data	2 (2.2)	1 (2.0)	-	1 (10.0)	
**Uveitis**					
Yes	20 (20.4)	11 (21.6)	7 (20.6)	2 (20.0)	0 (0.0)
No	78 (79.6)	40 (78.4)	27 (79.4)	8 (80.0)	3 (100.0)
**TMJ-affected**					
Yes	26 (26.5)	14 (27.5)	11 (32.4)	1 (10.0)	0 (0.0)
No	72 (73.5)	37 (72.5)	23 (67.6)	9 (90.0)	3 (100.0)
ESR (mm/1sth), Median (IQR) ^5^					
A	12.0 (5.0–31.0)	9.0 (5.0–30.0)	23.0 (8.8–33.3)	5.0 (2.0–8.0)	13.0 (9.0–)
B	5.0 (2.0–8.0)	5.0 (2.0–9.0)	5.0 (2.0–7.0)	2.0 (2.0–5.0)	9.0 (5.0–)
C	5.0 (2.0–7.0)	5.0 (2.0–5.0)	5.0 (2.0–8.0)	2.0 (2.0–14.0)	-
D	5.0 (2.0–7.8)	5.0 (2.0–7.3)	5.0 (2.0–8.0)	4.0 (2.0–6.0)	7.0 (7.0–)
CRP (mg/L), Median (IQR) ^5^					
A	1.0 (1.0–7.8)	1.0 (1.0–5.4)	4.0 (1.0–23.3)	1.0 (1.0–4.0)	1.0 (1.0–)
B	1.0 (1.0–1.0)	1.0 (1.0–1.0)	1.0 (1.0–1.0)	1.0 (1.0–1.0)	1.0 (1.0–1.0)
C	1.0 (1.0–1.0)	1.0 (1.0–1.0)	1.0 (1.0–1.0)	1.0 (1.0–3.5)	-
D	1.0 (1.0–1.0)	1.0 (1.0–1.0)	1.0 (1.0–1.0)	1.0 (1.0–1.5)	1.0 (1.0–)
Joint count (Median, min–max) ^6^					
A	3.0 (0–44)	2 (1–4)	5.5 (1–44)	1 (1–3)	1.0 (0–2)
B	0.0 (0–7)	0 (0–7)	0 (0–7)	0 (1–3)	0.0 (0)
C	0.0 (0–4)	0 (0–4)	0 (0–3)	0 (0–2)	-
D	0.0 (0–11)	0 (0–4)	0 (0–11)	0 (0–2)	0.0 (0–2)
JADAS 10–CH (Median, IQR) ^6^					
A	8.3 (6.2–13.3)	7.5 (6.2–9.8)	13.9 (8.3–22.1)	4.8 (4.1–5.9)	3.9 (2.0–)
B	2.1 (0.6–4.2)	2.0 (0.7–3.9)	2.4 (0.1–4.2)	3.3 (0.8–6.0)	3.5 (0.0–)
C	0.5 (0.0–1.9)	0.5 (0.0–2.1)	1.0 (0.0–1.7)	0.1 (0.0–5.1)	-
D	0.9 (0.0–5.4)	0.3 (0.0–3.9)	0.8 (0.0–5.8)	2.9 (0.4–8.3)	2.3 (0.0–)
PGA ^6^					
A	3 (1.0–4.0)	2 (1.0–3.0)	5.0 (3.0–7.0)	1.0 (1.0–2.0)	2.0 (1.0–)
B	0 (0.0–1.0)	0.5 (0.0–1.0)	0.0 (0.0–1.4)	0.5 (0.0–1.0)	0.0 (0.0–0.0)
C	0 (0.0–1.0)	0.0 (0.0–0.5)	0.0 (0.0–0.0)	0.0 (0.0–0.3)	-
D	0 (0.0–1.0)	0.0 (0.0–0.0)	0.0 (0.0–2.0)	0.5 (0.0–1.3)	0.0 (0.0–)
CHAQ ^6^					
A	0.5 (0.1–1.1)	0.4 (0.1–0.8)	0.8 (0.1–1.4)	0.3 (0.1–0.5)	0.1 (0.1–0.1)
B	0.1 (0.0–0.6)	0.1 (0.0–0.5)	0.3 (0.0–1.1)	0.1 (0.0–0.2)	0.6 (0.0–)
C	0.0 (0.0–0.3)	0.0 (0.0–0.3)	0.1 (0.0–0.4)	0.1 (0.0–0.2)	-
D	0.0 (0.0–0.4)	0.0 (0.0–0.4)	0.0 (0.0–0.5)	0.2 (0.0–0.4)	0.1 (0.0–)

The data are presented as numbers (*n*) of children and valid percentages (%). The missing data are presented as numbers and percentages from the total number of patients. ^1^ Persistent Oligoarthritis, *n* = 44 (44.9%) and extended Oligoarthritis, *n* = 7 (7.1%).^2^ Includes diagnoses of psoriasis-related juvenile idiopathic arthritis (ICD-10: M09.0*, L40.5), *n* = 1 and juvenile arthritis, unspecified (ICD-10: M08.9), *n* = 2. ^3^ Medication at time of sampling. ^4^ Disease activity was determined as described earlier and presented here at the time of diagnosis, 1 year after, and time of sampling, respectively. ^5^ Laboratory values. ^6^ Disease activity parameters were recorded at three time points: A = at the time of diagnosis, B = one year after diagnosis, C = three years after diagnosis, and D = at the time of sampling. Abbreviations: HLA = human leukocyte antigen, ANA = antinuclear antibody, TMJ = temporomandibular joint, ESR = an erythrocyte sedimentation rate, CRP = c-reactive protein, JADAS = juvenile arthritis disease activity score, PGA = physician global assessment of disease activity, and CHAQ = child health assessment questionnaire (CHAQ).

**Table 2 ijms-25-08323-t002:** The distribution of IL-17A and IL-10 genotypes in the JIA patients and controls.

SNP	Controls ^1^, *n* = 99 (%)	Controls ^1,2^ *n* = 994 (%)	JIA Patients ^3^, *n* = 98 (%)	OR (95% Cl)	*p*-Value
** *IL-17A* **					
**rs9395767**					
AA	31 (32.3)		27 (27.6)		0.749
AT	43 (43.4)		42 (42.9)		
TT	25 (25.3)		29 (29.6)		
AT + TT ^4^	67 (67.7)		71 (72.4)	1.26 (0.68–2.32)	0.534
A-allele	105 (53.0)		96 (49.0)	1.18 (0.79–1.75)	0.394
T-allele	93 (47.0)		100 (51.0)		
HWE (*p*-value)	0.211		0.158		
**rs4711998**					
GG	47 (47.5)		41 (41.8)		0.521
AG	45 (45.5)		46 (46.9)		
AA	7 (7.1)		11 (11.2)		
AG + AA ^4^	52 (52.2)		57 (58.2)	1.26 (0.72–2.21)	0.475
G-allele	139 (70.2)		128 (65.3)	1.25 (0.82–1.91)	0.299
A-allele	59 (29.8)		68 (34.7)		
HWE (*p*-value)	0.390		0.723		
**rs8193036**					
TT	43 (43.0)		28 (28.6)		0.089
CT	40 (40.4)		48 (49.0)		
CC	19 (19.2)		22 (22.4)		
CT + CC ^4^	59 (59.6)		70 (71.4)	1.92 (1.06–3.47)	**0.038**
T-allele	126 (63.6)		104 (53.1)	1.43 (0.96–2.13)	0.079
C-allele	78 (36.4)		92 (46.9)		
HWE (*p*-value)	0.087		0.869		
**rs2275913**					
GG		341 (34.3)	24 (24.5)		0.246
GA		464 (46.7)	47 (48.5)		
AA		189 (19.0)	27 (27.6)		
GA + AA ^4^		653 (65.7)	74 (75.5)	1.61 (0.99–2.60)	**0.049 ***
G-allele		1146 (57.6)	95 (48.5)	1.45 (1.08–2.60)	**0.014 ***
A-allele		842 (42.4)	101 (51.5)		
HWE (*p*-value)		0.165	0.692		
** *IL-10* **					
**rs1800896**					
TT		318 (32.5)	22 (22.4)		0.076
TC		461 (47.1)	57 (58.2)		
CC		200 (20.4)	19 (20.9)		
TC + CC ^4^		661 (67.5)	76 (76.9)	1.66 (1.02–2.72)	**0.042 ***
T-allele		1097 (56.0)	101 (51.5)	1.20 (0.89–1.61)	0.228
C-allele		861 (44.0)	95 (48.5)		
HWE (*p*-value)		0.165	0.104		
**rs1800871**					
GG		594 (60.4)	58 (59.2)		0.076
GA		329 (33.5)	39 (39.8)		
AA		60 (6.1)	1 (1.0)		
GA + AA ^3^		40 (40.4)	36 (39.6)	1.06 (0.69–1.61)	0.810
G-allele		1517 (76.3)	145 (74.0)	0.86 (0.59–1.26)	0.424
A-allele		449 (23.7)	37 (26.0)		
HWE (*p*-value)		0.114	0.045		
**rs1800872**					
GG		594 (60.4)	55 (60.4)		0.076
GT		329 (33.5)	35 (38.5)		
TT		60 (6.1)	1 (1.1)		
GT + TT ^3^		40 (40.4)	36 (39.6)	1.06 (0.69–1.61)	0.810
G-allele		1517 (76.3)	145 (74.0)	0.86 (0.59–1.26)	0.424
T-allele		449 (23.7)	37 (26.0)		
HWE (*p*-value)		0.114	0.045		

The data are presented as numbers (*n*) of children and valid percentages (%). Two control groups are included: ^1^ Genome 1000 FIN population [31] and ^2^ Finnish STEPS study cohort [32]. Of the latter, data of *IL-17A* rs2275913 and *IL-10* rs1800896 and rs1800872 are available.^3^ Includes diagnoses of juvenile arthritis subtypes: oligoarthritis, polyarthritis, juvenile enthesitis-related arthritis, juvenile arthritis psoriasis, and juvenile arthritis unspecified. Fisher’s exact tests were used to analyze *p*-values. The odds ratio (OR) and the corresponding 95% confidence intervals (95% CI) were calculated. ^4^ Dominant model where wildtype was compared with heterozygous and homozygote variants. *p*-values < 0.05 were considered significant (*).

**Table 3 ijms-25-08323-t003:** The distribution of *IL-17A* and *IL-10* genotypes in JIA patients by three main JIA subtypes.

SNPs	Controls ^1^, *n* = 99 (%)	Controls ^1,2^ *n* = 994 (%)	Oligoarthritis, *n* = 51 (%)	OR (95% Cl), *p*-Value	Polyarthritis, *n* = 34 (%)	OR (95% Cl), *p*-Value	ERA, *n* = 10 (%)	OR (95% Cl), *p*-Value
** *IL-17A* **								
rs9395767								
AA	31 (31.3)		18 (35.3)	0.858	7 (20.6)	0.457	1 (10.0)	0.327
AT	43 (43.4)		20 (39.2)		16 (47.1)		5 (50.0)	
TT	25 (25.2)		13 (25.5)		11 (32.4)		4 (40.0)	
AT + TT ^3^	68 (68.7)		33 (64.7)	0.88 (0.43–1.79), 0.718	27 (79.4)	1.84 (0.73–4.68), 0.275	9 (85.7)	4.30 (0.52–35.40), 0.277
A-allele	105 (53.0)		56 (54.9)	0.93 (0.57–1.50), 0.758	30 (44.1)	1.43 (0.82–2.50), 0.206	7 (0.35)	2.10 (0.80–5.48), 0.131
T-allele	93 (47.0)		46 (45.1)		38 (55.9)		13 (0.65)	
rs4711998								
GG	47 (47.5)		22 (43.1)	0.608	13 (38.2)	0.643	4 (40.0)	0.367
AG	45 (45.5)		23 (45.1)		18 (52.9)		4 (40.0)	
AA	7 (7.1)		6 (11.8)		3 (8.8)		2 (20.0)	
AG + AA ^3^	52 (52.5)		28 (56.0)	1.19 (0.603–2.35), 0.730	21 (61.8)	1.46 (0.66–3.24), 0.426	6 (60.0)	1.36 (0.36–5.10), 0.748
G-allele	139 (70.2)		67 (65.7)	1.23 (0.74–2.05), 0.798	44 (64.7)	1.29 (0.72–2.30), 0.843	12 (60.0)	1.57 (0.61–4.04), 0.349
A-allele	59 (29.8)		35 (34.3)		24 (35.3)		8 (40.0)	
rs8193036								
TT	40 (40.4)		15 (29.4)	0.219	10 (29.4)	0.354	3 (30.0)	0.514
CT	40 (40.4)		24 (47.1)		17 (50.0)		6 (60.0)	
CC	19 (16.2)		12 (23.5)		7 (20.6)		1 (10.0)	
CT + CC ^3^	59 (59.6)		34 (68.0)	1.84 (0.90–3.79), 0.113 ^4^	24 (70.6)	1.84 (0.80–4.26), 0.162 ^2^	7 (70.0)	1.79 (0.44–7.34), 0.514
T-allele	126 (63.6)		54 (52.9)	1.44 (0.89–2.32), 0.140	37 (54.4)	1.93 (1.11–3.36), **0.020 ***	12 (60.0)	1.08 (0.42–2.75), 0.155
C-allele	78 (36.4)		48 (47.1)		31 (45.6)		8 (40.0)	
rs2275913								
GG		341 (34.3)	9 (17.6)	**0.046** ^4^	8 (23.5)	0.055	5 (50.0)	0.521
GA		464 (46.7)	29 (56.9)		14 (41.2)		3 (30.0)	
AA		189 (19.0)	13 (25.5)		12 (35.3)		2 (20.0)	
GA + AA ^3^		653 (65.7)	42 (82.4)	2.44 (1.17–5.07), **0.014**	26 (76.5)	1.70 (0.66–3.98), 0.192	5 (50.0)	0.52 (0.15–1.09), 0.299
G-allele		1146 (57.6)	47 (46.1)	1.59 (1.07–2.37), **0.022** ^5^	30 (44.1)	1.72 (1.06–2.81), 0.028 ^3^	13 (65.0)	0.73 (0.29–1.87), 0.509
A-allele		842 (42.4)	55 (53.9)		38 (55.9)		7 (35.0)	
** *IL-10* **								
**rs1800896**								
TT		318 (32.5)	11 (21.6)	0.200	8 (23.5)	0.387	1 (10.0)	0.266
TC		461 (47.1)	30 (58.5)		20 (58.8)		7 (70.0)	
CC		200 (20.4)	10 (19.6)		6 (17.6)		2 (20.0)	
TC + CC ^3^		661 (67.5)	40 (78.4)	1.75 (0.89–3.46), 0.103	26 (76.5)	1.56 (0.70–3.49), 0.289	(90.0)	4.33 (0.55–34.32), 0.130
T-allele		1097 (56.0)	52 (51.0)	1.23 (0.82–1.82), 0.318	36 (52.9)	1.20 (0.74–2.33), 0.471	9 (45.0)	1.64 (0.68 –3.98), 0.272
C-allele		861 (44.0)	50 (49.0)		32 (47.1)		11 (55.0)	
**rs1800871**								
GG		594 (60.4)	30 (58.8)	0.136	19 (55.9)	0.538	8 (80.0)	0.412
GA		329 (33.5)	21 (41.2)		14 (41.2)		2 (20.0)	
AA		60 (6.1)	0 (0.0)		1 (2.9)		2 (20.0)	
GA + AA ^3^		40 (40.4)	21 (41.2)	1.03 (0.60–1.90), 0.819	15 (44.1)	1.20 (0.61–2.40), 0.894	1 (14.4)	0.38 (0.08–1.81), 0.208
G-allele		1517 (76.3)	81 (79.4)	0.88 (0.54–1.43), 0.597	52 (76.5)	0.99 (0.52–1.89), 0.972	18 (90.0)	1.13 (0.44–2.85), 0.802
A-allele		449 (23.7)	21 (20.6)		16 (23.5)		6 (10.0)	

Data are presented as numbers (*n*) of children and valid percentages (%). Fisher’s exact tests or Chi-square were used to analyze *p*-values. The odds ratio (OR) and the corresponding 95% confidence intervals (95% CI) were calculated. Two control groups are included: ^1^ Genome 1000 FIN population [31] and ^2^ Finnish STEPS study cohort [32]. Of the latter, data of *IL-17A* rs2275913 and *IL-10* rs1800896 and rs1800872 are available. ^3^ Dominant model where wildtype was compared with heterozygous and homozygote variants. *p*-values < 0.05 were considered significant (*). ^4^ When oligo- and polyarthritis groups are combined (*n* = 85), the *p*-value for rs2275913 is 0.007 (OR 211; 95% Cl 1.2–3.60) and for rs8193036, it is 0.049 (OR 1.84; 95% Cl 1.84–1.00). In the combined group, *p*-values and ORs at allelic level analyses were for ^5^ *IL-17A* rs2275913 A-allele 0.0020 OR 1.64; 95% C1 1.20–2.25.

**Table 4 ijms-25-08323-t004:** Association of IL-17A and IL-10 polymorphisms with ANA, HLA-B27, uveitis and TMJ arthritis.

SNPs	ANA (%)		OR (95% Cl), *p*-Value	Uveitis (%)		OR (95% Cl), *p*-Value	TMJ (%)		OR (95% Cl), *p*-Value
	Positive	Negative		Yes	No		Yes	No	
**ALL**	65 (67.7)	31 (32.3)		20 (20.4)	78 (79.6)		26 (26.5)	72 (73.5)	
Missing cases	2 (2.0)			**-**			**-**		
** *IL-17A* **									
rs9395767									
AA	15 (55.6)	12 (44.4)	0.084 ^2^	4 (14.8)	23 (85.2)	0.220	5 (18.5)	22 (81.5)	0.400
AT	26 (65.0)	14 (35.0)		12 (28.6)	30 (71.4)		11 (26.2)	31 (73.8)	
TT	24 (82.8)	5 (17.2)		4 (13.8)	25 (86.2)		10 (34.5)	19 (65.5)	
AT + TT ^1^	50 (72.5)	19 (27.5)	2.11 (0.84–5.31), 0.111	16 (22.5)	55 (75.5)	1.52 (0.56–4.14), 0.576	21 (29.6)	50 (70.4)	1.60 (0.18–1.62), 0.268
rs4711998									
GG	27 (67.5)	13 (32.5)	0.255	7 (17.1)	34 (82.9)	0.363	11 (26.8)	30 (73.2)	0.792
AG	29 (63.0)	17 (37)		9 (19.6)	37 (80.4)		13 (28.3)	33 (71.7)	
AA	9 (90.0)	1 (10.0)		4 (36.4)	7 (63.6)		2 (18.2)	9 (81.8)	
AG + AA ^1^	38 (67.9)	18 (32.1)	1.02 (0.43–2.42), 0.971	13 (22.8)	44 (77.2)	1.34 (0.59–3.05), 0.487	15 (26.3)	42 (73.7)	0.98 (0.50–1.91), 0.955
rs8193036									
TT	16 (57.1)	12 (42.9)	0.324	7 (25.0)	21 (75.0)	0.610	5 (17.9)	23 (82.1)	0.149
CT	33 (70.2)	14 (29.8)		10 (20.8)	38 (79.2)		17 (35.4)	31 (64.6)	
CC	16 (76.2)	5 (23.8)		3 (13.6)	19 (86.4)		4 (18.2)	18 (81.8)	
CT + CC ^1^	49 (72.1)	19 (27.9)	1.93 (0.77–4.84), 0.155	13 (18.6)	57 (81.4)	0.74 (0.33–1.67), 0.476	21 (30.0)	49 (70.0)	1.68 (0.70–4.02), 0.219
rs2275913									
GG	16 (66.7)	8 (33.3)	0.979	4 (16.7)	20 (83.3)	0.684	6 (25.0)	18 (75.0)	0.763
GA	31 (67.4)	15 (32.6)		9 (19.9)	38 (80.9)		14 (29.8)	33 (70.2)	
AA	18 (69.2)	8 (30.8)		7 (25.7)	20 (74.1)		6 (22.0)	21 (77.8)	
GA + AA ^1^	49 (68.1)	23 (31.9)	1.07 (0.40–2.85), 0.999	16 (21.6)	58 (78.4)	1.30 (0.22–2.43), 0.601	20 (27.0)	54 (73.0)	1.08 (0.49–2.38), 0.845
** *IL-10* **									
rs1800896									
TT	15 (68.2)	7 (31.8)	0.891	4 (18.2)	18 (81.8)	0.958	8 (36.4)	14 (63.6)	0.161
TC	38 (69.1)	17 (30.9)		12 (21.1)	45 (78.9)		16 (28.1)	41 (71.9)	
CC	12 (63.2)	7 (36.8)		4 (21.1)	15 (78.9)		2 (10.5)	17 (89.5)	
TC + CC ^1^	50 (67.6)	24 (32.4)	0.97 (0.35–2.70), 0.957	16 (21.1)	60 (78.9)	1.17 (0.43–3.11), 0.769	18 (23.7)	58 (76.3)	0.65 (0.33–1.29), 0.236
rs1800871									
GG	36 (63.2)	21 (36.8)	0.441	12 (20.7)	46 (79.3)	0.878	15 (57.7)	43 (74.1)	0.247
GA	28 (73.7)	10 (26.3)		8 (20.5)	31 (79.5)		10 (38.5)	29 (74.4)	
AA	1 (100.0)	0 (0.0)		0 (0.0)	1 (100.0)		1 (100.0)	0 (0.0)	
GA + AA ^1^	29 (74.4)	10 (25.6)	1.69 (0.69–4.15), 0.249	8 (20.0)	32 (80.0)	0.97 (0.44–2.15), 0.934	11 (27.5)	29 (72.5)	1.06 (0.55–2.07), 0.857

The data are presented as numbers (*n*) of children and valid percentages (%). Fisher’s exact tests or Chi-square were used to analyze *p*-values. The odds ratio (OR) or relative risk (RR) and the corresponding 95% confidence intervals (95% CI) were calculated. ^1^ Dominant model where wildtype was compared with heterozygous and homozygote variants. ^2^ When major genotype AA was compared with homozygote variant genotype TT: OR 3.8; 95% Cl 1.13–13.09.

**Table 5 ijms-25-08323-t005:** Serum cytokine levels in the patients.

Subjects	IL-17A ^1^	IL-17F ^1^	IL-10 ^1^	IL-6 ^1^
All (*n* = 98)	1.1 (55.4)	0.20 (0.78)	7.57 (47.82)	131.4 (1953.75)
**Sex**				
Male (*n* = 27)	1.1 (85.8)	0.19 (0.94)	22.61 (49.39)	193.37 (859.79)
Female (*n* = 71)	1.1 (50.1)	0.20 (0.66)	5.11 (44.74)	111.92 (2235.0)
*p*-value	0.943	0.501	0.093	0.913
**Subtype**				
Oligoarthritis (*n* = 51)	1.1 (0.0)	0.16 (0.45)	7.01 (30.72)	39.06 (2235.00)
Polyarthritis (*n* = 34)	1.1 (231.33)	0.37 (1.15)	8.61 (75.53)	264.24 (9706.65)
ERA (*n* = 10)	1.1 (138.48)	0.19 (0.50)	30.87 (51.27)	38.27 (20.49)
*p*-value	0.227	0.348	0.204	0.259
**Activity ^2^**				
Inactive (*n* = 42)	1.10 (35.02)	0.30 (0.78)	2.63 (29.63)	69.88 (1555.32)
Low (*n* = 7)	1.10 (85.80)	0.33 (2.32)	18.84 (94.81)	388.65 (3368.15)
Median (*n* = 14)	1.10 (174.03)	0.31 (6.03)	53.00 (98.02)	195.58 (10650.24)
High (*n* = 16)				
*p*-value	0.742	**0.012**	**0.006**	0.549

^1^ All cytokine concentrations (median and interquartile range, IQR) in the serum are expressed as pg/mL. ^2^ Disease activity at the time of sampling includes the patients with oligoarthritis or polyarthritis and is based on the JADAS10 score, as defined by Consolaro et al. [33,34].

**Table 6 ijms-25-08323-t006:** Association of studied *IL-17A* and *IL-10* polymorphisms with serum IL-17A, IL-17F, IL-10 and IL-6.

SNP	Number of Subjects	IL-17A ^1^	IL-17F ^1^	IL-10 ^1^	IL-6 ^1^
** *IL-17A* **					
rs9395767					
AA	27	1.10 (50.10)	0.17 (0.45)	0.15 (47.3)	838.58 (11445.0)
AT	42	1.10 (27.07)	0.18 (0.48)	6.0 (30.29)	64.33 (1283.75)
TT	29	1.1 (277.25)	0.36 (2.31)	22.72 (96.72)	134.23 (728.55)
*p*-value		0.293	**0.030 ^2^**	0.088	0.243
rs4711998					
GG	41	1.10 (0.00)	0.11 (0.30)	0.15 (30.29)	111.92 (1595.21)
AG	46	1.10 (246.24)	0.33 (0.87)	11.39 (52.42)	187.39 (3539.56)
AA	11	1.10 (89.39)	0.26 (1.61)	6.83 (94.81)	67.95 (3912.00)
*p*-value		0.063 ^1^	**0.010 ^3^**	0.337	0.910
rs8193036					
TT	28	1.10 (22.49)	0.13 (0.68)	4.05 (29.70)	120.46 (1595.00)
CT	48	1.10 (82.19)	0.24 (1.58)	6.92 (77.77)	47.32 (1451.57)
CC	22	1.10 (167.55)	0.20 (0.36)	14.90 (48.21)	316.71 (10693.61)
*p*-value		0.390	0.248	0.563	0.287
rs2275913					
GG	24	1.10 (147.43)	0.36 (0.90)	9.50 (42.31)	28.76 (236.69)
AG	47	1.10 (50.10)	0.21 (0.68)	10.24 (50.92)	347.93 (1860.00)
AA	27	1.10 (45.68)	0.12 (0.82)	6.64 (56.90)	133.78 (9685.00)
*p*-value		0.881	0.123	0.776	0.277
** *IL-10* **					
rs1800896					
TT	22	6.80 (197.69)	0.54 (3.09)	37.88 (110.90)	243.65 (9923.23)
TC	57	1.10 (28.28)	0.19 (0.48)	6.83 (31.59)	67.95 (1562.00)
CC	19	1.10 (85.80)	0.07 (0.36)	0.15 (142.02)	337.71 (1860.00)
*p*-value		0.189	**0.021 ^4^**	0.125	0.406
rs1800871					
GG	58	1.10 (86.70)	0.19 (1.48)	12.69 (58.71)	131.39 (1578.82)
GA	39	1.10 (29.99)	0.21 (0.76)	6.83 (22.46)	108.64 (6605.00)
AA	1				
*p*-value		0.735	0.673	0.307	0.873

^1^ All cytokine concentrations (median and interquartile range, IQR) in the serum are expressed as pg/mL. With the dominant model, *p*-values are 0.019 for ^2^ IL-17A, 0.002 for ^3^ IL-17F, and 0.01 for ^4^ IL-10.

**Table 7 ijms-25-08323-t007:** Primers used for PCR and Sanger sequencing.

Target	Primers	Annealing Temperature (°C)	PCR Product Size (bp)
** *IL-17A* **			
rs147789086 ^2^	F: 5′-TGTGCACTAGGTGGAGGAAAC-3′ R: 5′-CCAGCACACAAGGAGCAATG-3′ ^1^	60	724
rs9395767			
rs2275913	F: 5′-TGACCCATAGCATAGCAGCTC-3′ ^1^ R: 5′-TCTTGTGTGGTTTAGCCCCAA-3′	60	576
rs8193037 ^2^			
rs3819025 ^2^			
rs8193038 ^2^			
rs8193036	F: 5′-GGGCGGAGAAGGGTGACATA-3′ ^1^ R: 5′-GATTCCTTGGCCAGGTGTTGT-3′	60	521
rs4711998			
** *IL-10* **			
rs1800896	F: 5′-CCAGATATCTGAAGAAGTCCTG-3′ R: 5′-CCTAGGTCACAGTGACGTGG-3′ ^1^	55	901
rs1800871			
rs1800872			

^1^ Primer used for Sanger sequencing. ^2^ The frequency of the point mutation was less than 5%, so it is not presented in Table 2 or included in the analyses.

## Data Availability

Data are available upon request.

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
