# Peer review of "Association of IL-17A and IL-10 Polymorphisms with Juvenile Idiopathic Arthritis in Finnish Children"

_ijms, 2024, doi:10.3390/ijms25158323_

Round 1
Reviewer 1 Report
Comments and Suggestions for Authors
Although the study was of modest interest, I have the following comments:
1) The INTRODUCTION part was too lengthy and it could be condensed to enhance its readability and focus. I have some specific suggestions:
- summarize key points about (JIA) and its subtypes without going into excessive detail. Focus on the most relevant aspects for understanding the study’s context.
- focus on key cytokines: Condense the description regarding the roles of various cytokines in JIA. It is enough to emphasize IL-17A and IL-10, as these are central to the study.
- remove redundant and unnecessary information: For example, it is not required to
describe the detailed role of Treg, as this is not the research focus in this study.
2) What is the unique contribution of this study ? Is this study a replication study ? or a novel study ?, This aspect should be pointed out in the introduction.
Author Response
- The INTRODUCTION part was too lengthy and it could be condensed to enhance its readability and focus. I have some specific suggestions:
- summarize key points about (JIA) and its subtypes without going into excessive detail. Focus on the most relevant aspects for understanding the study’s context.
- focus on key cytokines: Condense the description regarding the roles of various cytokines in JIA. It is enough to emphasize IL-17A and IL-10, as these are central to the study.
- remove redundant and unnecessary information: For example, it is not required to describe the detailed role of Treg, as this is not the research focus in this study.
Answer:
Thank you for the useful comments. In this revised version we have substantially condensed the the introduction section leaving out unessential information according to referee’s suggestions.
- What is the unique contribution of this study ? Is this study a replication study ? or a novel study ?, This aspect should be pointed out in the introduction.
Answer:
We concur that several SNPs in the IL-17 and IL-10 promoter regions have been individually studied in patients with JIA. However, to the best of our knowledge, this is the first study to comprehensively examine these seven SNPs together in JIA patients. To clarify the unique contribution of our research, we have added few sentences to the end of the introduction section (lines: 122-126).
Reviewer 2 Report
Comments and Suggestions for Authors
My comments to the authors are outlined below.
1. Conceptually, the study's goals and hypothesis are not well formulated, if at all. It is unclear what the main premise of the study is. Does the study aim to identify genetic biomarkers associated with JIA? All the authors have done is report a weak association between SNPs in IL-17 and IL-10 with various clinical phenotypes of JIA.
2. However, the reported results are not convincing due to numerous methodological insufficiencies. For example, the authors test 4 and 2 SNPs across IL-17 and IL-10, respectively. However, no correction for multiple testing is offered in these analyses, which renders these results indistinguishable from random in light of the weak associations.
3. Furthermore, the study, at least to this reviewer, seems like a fishing expedition. For instance, in Figure 1, the authors report different LD patterns between cases and controls. I would not put too much trust in these haplotype blocks, however. First, the sample size is very small to reflect any reported LD relationships between these groups accurately. Second, because of it, this can be completely stochastic. Did the authors attempt to investigate the LD relationships for these particular SNPs across the entire 1K genome for the EA subjects or an existing large GWAS in the Finn population?
4. The authors have attempted further to establish a link between the serum levels of IL-17 and IL-10 and different genotypes of the SNPs of interest. Again, only very weak correlations are observed. Additionally, the results in Table 6 are confusing, as it appears that the authors have tested all SNPs across the 2 cytokines, even though the rationale for this is not provided and the 2 cytokines are located on different chromosomes.
5. The authors provide information about PCR primers and conditions in the methods section but do not present results from these assays!
6. The authors need to provide more information about the multiplex immunoassays pertaining to the specificity and accuracy of the reported results.
Author Response
Reviewer 2
Conceptually, the study's goals and hypothesis are not well formulated, if at all. It is unclear what the main premise of the study is. Does the study aim to identify genetic biomarkers associated with JIA? All the authors have done is report a weak association between SNPs in IL-17 and IL-10 with various clinical phenotypes of JIA.
Answer:
We agree that the initial formulation of the goals and hypotheses lacked precision. This study aimed at recognizing biomarkers for JIA, its subgroups and clinical entities in a Finnish JIA population. We have now tried to clarify the rationale and goal of this study in the final paragraph of introduction (lines 122 – 126).
- However, the reported results are not convincing due to numerous methodological insufficiencies. For example, the authors test 4 and 2 SNPs across IL-17 and IL-10, respectively. However, no correction for multiple testing is offered in these analyses, which renders these results indistinguishable from random in light of the weak associations.
Answer:
We acknowledge that correction for multiple testing was not applied in the analyses presented in Table 2. This decision was made due to the limited sample size available in both the subject and control groups. Typically, such corrections are employed to avoid Type I errors when a large number of tests are conducted without pre-planned hypotheses, as in GWAS studies. However, this study was an exploratory study, aiming to determine whether specific IL-17A and IL-10 SNPs occur more frequently among JIA patients compared to the healthy population, and to investigate potential differences in prevalence among various JIA subtypes. In exploratory studies, especially with a limited number of subjects, applying Bonferroni corrections can be imprudent due to the increased risk of Type II errors.
Based on these findings, we can proceed with more detailed studies, incorporating appropriate corrections for multiple testing. For instance, in comparisons between cytokines and SNPs, we have applied the recommended corrections.
To enhance statistical power, we included another group of controls based on the STEPS study conducted among 923 healthy children in Turku, Finland in which data of IL-17A rs2275913 and IL-10 rs1800896 and rs1800872 are available. Consequently, we have revised the materials and methods section (lines: 455-457 and 508-510), Tables 2 and 3, the results section (lines: 162, 170,173, 221 and 223), and the discussion section (lines: 343-346) to reflect these changes.
- Furthermore, the study, at least to this reviewer, seems like a fishing expedition. For instance, in Figure 1, the authors report different LD patterns between cases and controls. I would not put too much trust in these haplotype blocks, however. First, the sample size is very small to reflect any reported LD relationships between these groups accurately. Second, because of it, this can be completely stochastic. Did the authors attempt to investigate the LD relationships for these particular SNPs across the entire 1K genome for the EA subjects or an existing large GWAS in the Finn population?
Answer:
Thank you for your valuable comment. In this study the purpose to include the LD figures was to visualize how the frequencies of the studied SNPs differ between control subjects and patients with JIA, as well as among various subtypes of JIA. Thus, our intention was not to investigate the LD relationships for these particular SNPs across the entire 1K genome. We acknowledge that the sample size is limited and that there are additional SNPs in the promoter regions of IL-17A and IL-10 which were not examined in this study. To address this limitation, we have added a sentence at the end of the discussion section (lines: 433-436).
- The authors have attempted further to establish a link between the serum levels of IL-17 and IL-10 and different genotypes of the SNPs of interest. Again, only very weak correlations are observed. Additionally, the results in Table 6 are confusing, as it appears that the authors have tested all SNPs across the 2 cytokines, even though the rationale for this is not provided and the 2 cytokines are located on different chromosomes.
Answer:
We thank you for the useful comment. All, IL-17A, IL-17F and IL-6 are proinflammatory cytokines which have been shown to play a significant role in JIA pathogenesis. Here we have selected IL-6 as an intrinsic control of cytokine production and a representative of the state of general inflammation.
IL-17 family comprises six structurally related members (IL-17A to IL-17F) and sequence homology is highest between IL-17A and IL-17F (55%). They can form homodimers and heterodimers to signal through the same receptor complex. That’s why we include IL-17F in the comparison.
IL-10 is a major anti-inflammatory cytokine and can inhibit the activation and effector functions of T cells. By measuring the cytokine, we may better understand the balance between pro- and anti-inflammatory cytokines. We understand the concern by this reviewer, a sentence to describe its limitation was included in the end of discussion (Lines 436-440).
- The authors provide information about PCR primers and conditions in the methods section but do not present results from these assays!
Answer:
The primers provided in the table 7 were used for PCR and Sanger sequencing of followed IL-17A and IL-10 SNPs: rs147789086, rs9395767, rs2275913, rs8193037, rs3819025, rs8193038, rs4711998, rs1800896, rs1800871 and rs1800872. Only those SNPs which minor allele frequency was >5% were included in the analyses, therefore the frequencies of IL-17A rs147789086, rs3819025 and rs8193038 were not reported. To clarify this, the footnote (2) added in the table 7.
- The authors need to provide more information about the multiplex immunoassays pertaining to the specificity and accuracy of the reported results.
Answer:
The kit used for multiplex immunoassay is high quality with no or negligible cross-reactivity between the antibodies for an analyte and any of the other analytes un the panel. Reported Intra-assay CVs were only 3%, 2%, 3% and 5%, and the Inter-assay CVs were 13%, 10%, 11% and 7% for IL-17A, IL-17F, IL-10 and IL-6, respectively. One sentence has been added to the method section to provide more information regarding the specificity and accuracy of the method (lines: 510 – 512).

Reviewer 3 Report
Comments and Suggestions for Authors
General Comments:
In this study, the authors demonstrate a link between IL-17A and IL-10 gene variations and JIA in Finnish youth. They also reported the association of serum IL-17F and IL-10 levels with disease severity in their JIA cohort.
Specific comments
1. There is insufficient information about the control group used for comparison and the control group may not be suitable due to a lack of matching for age and sex with the JIA patients.
2. It will be interesting to compare the haplotype frequencies of the four IL17A SNPs and the three IL10 SNPs between control and patient groups.
3. The authors investigated the association of IL-17A and IL-10 SNPs with other cytokines including IL-17F and IL-6, what is the rationale behind this?
Author Response
Reviewer 3
General Comments:
In this study, the authors demonstrate a link between IL-17A and IL-10 gene variations and JIA in Finnish youth. They also reported the association of serum IL-17F and IL-10 levels with disease severity in their JIA cohort.
Specific comments:
- There is insufficient information about the control group used for comparison and the control group may not be suitable due to a lack of matching for age and sex with the JIA patients.
Answer:
We concur that ideally, the control group should be age- and gender-matched. However, given that the SNPs under study are not located on the X or Y chromosomes and occur equally in both sexes, and in addition there is no evidence suggesting that the genetics of the selected genes change with age. To substantiate this, we conducted additional analyses in both groups and no significant sex-based differences were observed concerning the prevalence of SNPs (all P values = >0.05). Two sentences have been added in the results section (lines: 153 – 155).
- It will be interesting to compare the haplotype frequencies of the four IL17A SNPs and the three IL10 SNPs between control and patient groups.
Answer:
We agree that it is important to compare the haplotype frequencies of the four IL17A SNPs and the three IL10 SNPs between control and patient groups. Actually this information was included in the manuscript (please see the result section 2.4 as well as Figures 1 and 2).
- The authors investigated the association of IL-17A and IL-10 SNPs with other cytokines including IL-17F and IL-6, what is the rationale behind this?
Answer:
IL-17A and IL-17F are produced by cells of the innate and adaptative immune system and activate the production of inflammatory mediators such as tumor necrosis factor α (TNFα), IL-1β, IL-6. Here we have selected IL-6 as an intrinsic control of cytokine production and a representative of the state of general inflammation. It is known that patients with active JIA should have higher IL-6 levels in their serum than those who have inactive disease.
What comes to the genetics, it is known that IL-17 family comprises six structurally related members (IL-17A to IL-17F) and sequence homology is highest between IL-17A and IL-17F (55%). They can form homodimers and heterodimers to signal through the same receptor complex. That’s why we include IL-17F in the comparison.

Round 2
Reviewer 1 Report
Comments and Suggestions for Authors
The manuscript was revised well.
Author Response
Answer: We thank this reviewer for the positive comment.
Reviewer 2 Report
Comments and Suggestions for Authors
I appreciate the authors' responses. However, this reviewer is not convinced that the reported results support the conclusions.
1. I disagree with the authors' premise that because their study is small, underpowered, and exploratory with fewer tests, they should not apply the Bonferroni test correction. This is a critical issue that should not be overlooked. First, this is exactly the case when correction for multiple testing (whether Bonferroni or another less stringent approach) should be applied. While I understand the desire to minimize Type II errors, Type I errors are far more detrimental. Second, there is a clear positive relationship between effect size and a p-value, with larger effect sizes associated with lower p-values. In fact, for studies with sufficiently large effect sizes, the Bonferroni correction will make little difference. However, this study's effect sizes seem pretty small, which necessitates larger sample sizes that are not presented in this study. Third, at the minimum, the authors should have reported power analysis to provide a certain measure of confidence that their results are not misleading.
2. The authors are encouraged to report their findings in light of what they are: potentially interesting, however, severely underpowered, and as a consequence, the reported associations need to be further examined in independent and hopefully much larger studies. Nowhere in the manuscript, the authors have stated that their findings are exploratory, and therefore, they need to be replicated before more credence is given to these results.
Comments on the Quality of English LanguageEnglish is fine
Author Response
I appreciate the authors' responses. However, this reviewer is not convinced that the reported results support the conclusions. I disagree with the authors' premise that because their study is small, underpowered, and exploratory with fewer tests, they should not apply the Bonferroni test correction. This is a critical issue that should not be overlooked. First, this is exactly the case when correction for multiple testing (whether Bonferroni or another less stringent approach) should be applied. While I understand the desire to minimize Type II errors, Type I errors are far more detrimental. Second, there is a clear positive relationship between effect size and a p-value, with larger effect sizes associated with lower p-values. In fact, for studies with sufficiently large effect sizes, the Bonferroni correction will make little difference. However, this study's effect sizes seem pretty small, which necessitates larger sample sizes that are not presented in this study. Third, at the minimum, the authors should have reported power analysis to provide a certain measure of confidence that their results are not misleading. Answer: We thank you for the constructive comments. We totally agree that both Type I and Type II errors are important and should be considered. As suggested, we have conducted the Power calculations for individual SNPs by using the QUANTO program, unfortunately we were not able to do power calculations for all SNPs together. When the case – control population rate is 1:10, the power of the study is relatively good (35 – 93%), but when the ratio is 1:1, definitely more cases (> n = 350) are needed to reach 80% study power. To address these issues and limitations, several sentences have been added in the material and methods part (lines: 507 – 513) and in the discussion (lines: 408 – 418). In addition, we have modified several words in lines 26, 138, 151, 152 and 304. 2. The authors are encouraged to report their findings in light of what they are: potentially interesting, however, severely underpowered, and as a consequence, the reported associations need to be further examined in independent and hopefully much larger studies. Nowhere in the manuscript, the authors have stated that their findings are exploratory, and therefore, they need to be replicated before more credence is given to these results. Answer: We fully agree. In order to state that our findings are exploratory, we have added a sentence in the end of abstract (lines: 27 - 28) and modified the discussions of limitations (lines 408 -418).
Reviewer 3 Report
Comments and Suggestions for Authors
All comments were adequately addressed.
Author Response

(The authors gave the same response as above.)

Round 3
Reviewer 2 Report
Comments and Suggestions for Authors
I have no further comments for the authors.
Comments on the Quality of English Languagenone